# Comparative Genomic Analysis of *PEBP* Genes in Cucurbits Explores the Interactors of Cucumber CsPEBPs Related to Flowering Time

**DOI:** 10.3390/ijms25073815

**Published:** 2024-03-29

**Authors:** Lianxue Fan, Ziyi Zhu, Xiaoru Lin, Xia Shen, Tianjiao Yang, Haixin Wang, Xiuyan Zhou

**Affiliations:** Key Laboratory of Biology and Genetic Improvement of Horticultural Crops (Northeast Region), Ministry of Agriculture, College of Horticulture and Landscape Architecture, Northeast Agricultural University, Harbin 150030, China; fanlianxue@neau.edu.cn (L.F.); a02220028@neau.edu.cn (Z.Z.); s230401045@neau.edu.cn (X.L.); s230402059@neau.edu.cn (X.S.); a02220036@neau.edu.cn (T.Y.); a02220039@neau.edu.cn (H.W.)

**Keywords:** cucurbit PEBP proteins, flowering regulation, gene expression analysis, growth and development processes

## Abstract

The family of phosphatidylethanolamine-binding proteins (PEBPs) participates in various plant biological processes, mainly flowering regulation and seed germination. In cucurbit crops, several *PEBP* genes have been recognized to be responsible for flowering time. However, the investigation of PEBP family members across the genomes of cucurbit species has not been reported, and their conservation and divergence in structure and function remain largely unclear. Herein, *PEBP* genes were identified from seven cucurbit crops and were used to perform a comparative genomics analysis. The cucurbit PEBP proteins could be classified into MFT, FT, TFL, and PEBP clades, and further, the TFL clade was divided into BFT-like, CEN-like, and TFL1-like subclades. The MFT-like, FT-like, and TFL-like proteins were clearly distinguished by a critical amino acid residue at the 85th position of the *Arabidopsis* FT protein. In gene expression analysis, *CsaPEBP1* was highly expressed in flowers, and its expression levels in females and males were 70.5 and 89.2 times higher, respectively, than those in leaves. *CsaPEBP5*, *CsaPEBP6*, and *CsaPEBP7* were specifically expressed in male flowers, with expression levels 58.1, 17.3, and 15.7 times higher, respectively, than those of leaves. At least five *CsaPEBP* genes exhibited the highest expression during the later stages of corolla opening. Through clustering of time-series-based RNA-seq data, several potential transcription factors (TFs) interacting with four CsaPEBPs were identified during cucumber corolla opening. Because of the tandem repeats of binding sites in promoters, NF-YB (Csa4G037610) and GATA (Csa7G64580) TFs appeared to be better able to regulate the *CsaPEBP2* and *CsaPEBP5* genes, respectively. This study would provide helpful information for further investigating the roles of *PEBP* genes and their interacting TFs in growth and development processes, such as flowering time regulation in cucurbit crops.

## 1. Introduction

Phosphatidylethanolamine-binding proteins (PEBPs) are widespread in plants and contain a conserved domain peculiar to eukaryotes, despite sharing an ancient functional unit with PEBPs in bacteria, archaea, and animals [1,2]. In the model plant *Arabidopsis*, AtPEBP proteins are mainly classified into three evolutionary branches, including FLOWERING LOCUS T (FT)-like, TERMINAL FLOWER1 (TFL1)-like, and MOTHER OF FT AND TFL1 (MFT)-like clades. In the AtFT-like clade, overexpression of both *FT* and *TWIN SISTER OF FT* (*TSF*) promotes earlier flowering [3]. Among the three members of the TFL1-like clade, *TFL1* is a major negative regulator of inflorescence meristem development that considerably delays flowering time, while *BROTHER OF FT AND TFL1* (*BFT*) play a role by acting redundantly with it [4]. Also, *ARABIDOPSIS THALIANA CENTRORADIALIS* (*ATC*), an additional paralogous of *TFL1*, exhibits a weak ability to complement the early flowering and terminal flower formation of *tfl1-1* mutant phenotypes, implying that *ATC* has a comparable function to *TFL1* and *BFT*. *MFT-like* genes, which are considered to be the evolutionary origin of *FT-like* and *TFL-like* clade genes, regulate the transcripts of several key genes during seed germination but also have a mild FT-like activity to encourage flowering in *Arabidopsis* [5,6]. In any case, PEBP family proteins are pivotal participants in controlling the plant flowering time.

FT and TFL1 proteins, with the opposite functions on flowering, have two highly conserved short motifs, DPDxP and GxHR, which presumably contribute to the conformation of the ligand binding pocket [1,7]. The amino acid residues encoded by the fourth exons of FT and TFL1 that are essential for conferring biological specificity on the two proteins can be divided into four segments (Segment A–D). Segment B is especially important for the determination of functional specificity between FT and TFL1, as is the adjacent segment C that contains the LYN/IYN triplet motif [8]. Additionally, the distal CCAAT box (TGTG(N2-3)ATG) and proximal CORE1/2 domains in the FT promoter region are found to be required cis-elements for binding by CONSTANS (CO) involved in photoperiodic pathways [9,10]. These findings offer valuable information for investigating FT- and TFL1-like specific interactors in other crops.

The Cucurbitaceae members (cucurbits) contain many important vegetable and fruit crops, such as cucumber (*Cucumis sativus* L.), melon (*Cucumis melo* L.), watermelon (*Citrullus lanatus* L.), pumpkin (*Cucurbita moschata* L.), cushaw (*Cucurbita argyrosperma* L.), zucchini (*Cucurbita pepo* L.), and winter squash (*Cucurbita maxima* Duch.), etc. [11]. The flowering time of the cucurbit crops is strongly correlated with early maturity and production, which is of great significance to the global or local economy [12]. To date, only a few flowering-related genes have been characterized in several cucurbit crops, including *CsFT* and *CsTFL1* in cucumber, *CmoFTL1* and *CmoFTL2* in Chinese pumpkin, and *FT* in watermelon, all of which belong to the PEBP family [13,14,15,16]. However, the regulatory mechanisms of the *FT* and *TFL-like* genes have not been well studied, and the roles of other PEBP members in regulating flowering time remain unclear.

The aim of this study was to identify the PEBP family genes in the seven cucurbit crops: cucumber, melon, watermelon, pumpkin, cushaw, zucchini, and winter squash. This investigation intends to comparatively analyze their chromosomal locations, phylogenetic relationships, and protein-conserved domains. Clustering of time-series-based RNA-seq data and promoter *cis*-element analyses were performed to explore the functions of the cucumber *PEBP* genes and their potential interacting transcription factors (TFs). These results could enhance our understanding of the cucurbit PEBP family members and provide available clues for delving deeper into their roles, especially in flowering time regulation.

## 2. Results

### 2.1. Identification of PEBP Family Genes in Seven Cucurbit Crops

Through mixtures of HMM with BLASTp searches, 65 *PEBP* genes were identified across seven cucurbit genomes, including 8 in cucumber (*CsaPEBP1*–*8*), 9 in melon (*CmePEBP1*–*9*), 7 in watermelon (*ClaPEBP1*–*7*), 10 in pumpkin (*CmoPEBP1*–*10*), 11 in cushaw (*CarPEBP1*–*11*), 10 in zucchini (*CpePEBP1*–*10*), and 10 in winter squash (*CmaPEBP1*–*10*) (Figure 1, Appendix A). The first three of them were diploid and contained fewer *PEBP* genes than the subsequent four cucurbits, which were tetraploid.

According to an intraspecies collinearity analysis, these PEBP members were shown to have a minority of collinear gene pairs, that is three in cucumber, two in melon, three in watermelon, three in pumpkin, five in cushaw, five in zucchini, as well as five in winter squash (Figure 1a–g, Appendix A). Moreover, only two pairs of tandem duplicates were determined in cushaw and pumpkin, including *CarPEBP9*–*CarPEBP10* and *CmoPEBP8*–*CmoPEBP9* (Figure 1a,f).

By mapping *ClaPEBP*s into the watermelon genome, seven genes were found to be dispersed over four of the eleven chromosomes (Chr02, Chr03, Chr04, and Chr10) (Figure 1a, Appendix A). Nine *CmePEBP*s were mapped on Chr00, Chr02, Chr04, Chr06, Chr08, Chr11, and Chr12 of the melon genome (Figure 1b). As shown in Figure 1c, eight *CsaPEBP* genes were highly concentrated on three chromosomes, especially the *CsaPEBP3*–*6* genes, which were localized on Chr03. *CarPEBP1*–*CarPEBP5*, *CarPEBP8* and *CarPEBP11* were each distributed on Chr03, Chr07, Chr10, Chr11, Chr13, Chr16, and Chr20, while both *CarPEBP6*–*7* and *CarPEBP9*–*10* were scattered on Chr14 and Chr18 of cushaw, respectively (Figure 1d, Appendix A). Similarly, the majority of the members of the *CmaPEBP*, *CmoPEBP*, and *CpePEBP* genes were dispersed across their chromosomes (Figure 1e–g, Appendix A).

### 2.2. Evolution of PEBP Genes in Arabidopsis and Seven Cucurbit Crops

To investigate the evolutionary relationships of the PEBP proteins, a comparative Neighbor-Joining (NJ) tree was conducted using the amino acid sequences encoded by the 72 *PEBP* genes from *Arabidopsis* and seven cucurbit crops (Figure 2, Appendix A). According to the NJ tree, a total of 72 PEBPs fall into four major branches, which were designated as MFT clade, FT clade, TFL clade, and PEBP clade. Further, the TFL clade could be divided into BFT-like, CEN-like, and TFL1-like subfamilies (Figure 2). Both MFT-like and TFL1-like subfamilies consisted of 17 cucurbit PEBP members, including at least two PEBPs from each cucurbit crop. The FT/TSF-like subfamily was comprised of one ClaPEBP, one CmePEBP, and one CsaPEBP, as well as two CarPEBPs, two CmaPEBPs, and two CmoPEBPs. The member number of the FT/TSF-like subfamily seemed to be consistent with the ploidy of different crops. The remaining BFT-like, CEN-like, and PEBP_bact-like subfamilies only had one PEBP protein from each of the seven cucurbits, except for the absence of any CpePEBP in the BFT-like subgroup (Figure 2).

A syntenic map illustrated the conserved syntenic blocks across the seven cucurbit crops, and 62 of the 65 *PEBP* genes were mapped to these orthologous blocks, indicating a high degree of macrosynteny (Figure 3, Appendix A). In total, 289 orthologous gene pairs were identified, with the number of genes matched varying from one to four. Among them, most one-to-one orthologous genes belonged to the *PEBP* genes from MFT-like, BFT-like, or PEBP-like subfamilies, e.g., *CmaPEBP5*/*6*/*8*/*9*, *CmoPEBP4*/*5*/*7*/*8*, and *CpePEBP2*/*4*/*6*/*10*. The one-to-four orthologous genes mainly come from the TFL1-like subfamily. The FT/TSF-like members matched one or two *PEBP*s as orthologous genes. Interestingly, *CmePEBP3*–*CsaPEBP6* seemed to be a unique orthologous gene pair (Figure 3, Appendix A), which likely resulted from an independent whole-genome duplication event between melon and cucumber.

Additionally, 28 orthologous gene pairs were identified between *Arabidopsis* and seven cucurbit crops, including five in cucumber, three in melon, five in watermelon, three in pumpkin, four in cushaw, five in zucchini, and three in winter squash (Appendix A). Except for *ClaPEBP1*–*AtATC*/*AtTFL* and *CsaPEBPE8*–*AtATC*/*AtTFL*, the rest were one-to-one orthologous gene pairs. All of these gene pairs fall into the MFT, TFL (BFT-like, CEN-like, and TFL-like), and PEBP clades, indicating that they might be derived from a common ancestor of *Arabidopsis* and seven cucurbit crops.

### 2.3. Comparative Analysis of the PEBP Family Gene Structure

Multiple sequence alignments showed that all the cucurbit PEBP proteins from the MFT clade, FT clade, and TFL clade had conserved PEBP domains and DPDxP motifs (Figure 4 and Appendix A). Also, the 85th amino acid ‘Y’ of AtFT, marked with a green rectangle, could clearly distinguish the MFT-like (W), FT-like (Y), and TFL-like (H) proteins, which was a key amino acid residue that determined FT-like and TFL-like functions. Most cucurbit FT-like and TFL-like proteins had amino acid residues marked with a red rectangle, indicating they might interact with the 14-3-3 protein. The other two amino acid sequences, ‘LGRQTVYAPGWRQN’ and ‘L/IYN’ were found to be highly conserved in the segments B and C of each cucurbit FT-like protein (Figure 4 and Appendix A), which were the critical motifs for determining the FT activity and the FT/TFL1 function. These results suggested that the FT-like clade from seven cucurbits was the most conserved, followed by the TFL-like clade, while the MFT-like clade was more variable.

In gene structure analysis, the 58 cucurbit *PEBP* genes from the MFT, FT, and TFL clades were identified to possess a similar number of exons, varying between three and six. Among them, approximately 75% (44) of *PEBP* genes had four exons (Appendix A). The PEBPs ranged from 233 bp to 6958 bp in gene length. From these, it could be inferred that the variations in the length and number of exons and introns may lead to the functional diversity of PEBP family members.

### 2.4. Cis-Acting Elements in the PEBP Promoter Regions

Within the putative promoter regions (2000 bp upstream from the transcription start site) of *PEBP* genes, there were five categories of *cis*-acting elements predicted, including the defensive or stress-related element, the developmental-related element, the hormone-responsive element, the light-responsive element, and the TF binding element (Figure 5, Appendix A). Among them, 40 of the 65 cucurbit PEBP promoters had no developmental-related element, indicating that they might be specific PEBP members involved in growth and development regulation. Exactly like the *CsaPEBP1* promoter, it was the only one of eight cucumber PEBP members with this element (CAT-box), which functioned in meristem expression (Appendix A). Moreover, the TF binding elements (e.g., AP-1, MYB, MYC, and HD-Zip) were identified in each PEBP promoter, accounting for almost 50% of the total (Figure 5, Appendix A), indicating that they might have the potential to regulate *PEBP* genes.

FT is a critical integration factor in the pathway of flowering initiation signal (florigen) in *Arabidopsis*, and the distal CCAAT box and proximal CORE domains in the promoter region are required for its proper expression [10]. Herein, we extracted the 7000-bp region of 11 *FT/TSF-like* genes from seven cucurbits and searched for the CCAAT box and CORE1/2 domain. Results showed that seven to sixteen CCACA boxes were present in all FT/TSF-like promoters, and the CORE1/2 domain was detected except for *CmePBEP1*, *CmoPEBP2*, and *CpePEBP8* (Appendix A). The above *cis*-acting regulatory elements would provide some information for the tissue-specific expression and interaction models of the cucurbit *PEBP* genes.

### 2.5. Spatio-Temporal Expression Patterns of CsaPEBP Genes

Cucumber is a typical model for unisexual flower development, and two *PEBP* family genes, *CsaPEBP2* (*CsFT*) and *CsaPEBP5* (*CsTFL1*), have been reported to influence flowering time [14,15]. To further investigate the roles of cucumber PEBP members, the expression patterns of eight *CsaPEBP*s were analyzed in various tissues (leaf, stem, tendril, male, female, and fruit). Reverse transcription-polymerase chain reaction (RT-qPCR) analysis indicated that each *CsaPEBP* gene displayed distinct expression patterns. In detail, *CsaPEBP2* and *CsaPEBP8* were expressed in all tissues (Figure 6b,h). *CsaPEBP1* was highly expressed in flowers, and its expression levels in females and males were 70.5 and 89.2 times higher, respectively, than those in leaves (Figure 6a). *CsaPEBP5*, *CsaPEBP6*, and *CsaPEBP7* were specifically expressed in male flowers, with expression levels 58.1, 17.3, and 15.7 times higher, respectively, than those of leaves (Figure 6e–g). *CsaPEBP3* and *CsaPEBP4* exhibited higher expression levels in the stem (Figure 6c,d). Furthermore, the expression levels of *CsaPEBP1*, *CsaPEBP2*, *CsaPEBP5*, *CsaPEBP6*, and *CsaPEBP7* were found to match the RNA-seq data reported by Li et al. [17], indicating that these genes could play stable roles during the growth and development of cucumber, especially in the development of floral organs.

In cucumber, the male or female corolla development could be divided into four stages, including green bud, green-yellow bud, yellow bud, and flowering based on the color and shape (Figure 7a). To explore the roles of several *CsaPEBP* genes in the process of male and female flower opening, RT-qPCR analysis was performed to examine their expression levels at each stage. As shown in Figure 7b, during the opening of the male flower, the expression levels of *CsaPEBP1* and *CsaPEBP7* had obviously increased at flowering (M4), *CsaPEBP5* and *CsaPEBP6* showed higher expression at the yellow bud (M3), while *CsaPEBP2* maintained a high expression level at both M3 and M4. In the progression of female corolla opening, the expression of *CsaPEBP1* and *CsaPEBP5* exhibited a similar, gradually upregulated trend, with a peak at flowering (C4), the peak expression of *CsaPEBP2* occurring at the yellow bud (C3), while *CsaPEBP6* and *CsaPEBP7* were expressed at a higher level at C3 and C4. Unfortunately, the expression patterns of the five *CsaPEBP* genes in the ovary did not exhibit sufficient regularity. These results suggested that *CsaPEBP1*, *CsaPEBP2*, *CsaPEBP5*, *CsaPEBP6*, and *CsaPEBP7* might be potential factors involved in regulating cucumber flowering, mainly functioning at the later stages: the yellow bud and flowering.

### 2.6. Screening for TFs That Interact with CsaPEBPs

FT or FT-like proteins, belonging to the PEBP family are well-known integrators in the flowering networks in multiple plants. Also, they can be directly activated by some TFs, or form complexes with TFs to trigger the onset of flowering [18]. In the current work, the clustering of time-series analysis between 1251 differentially expressed TFs and 4 *CsaPEBP* genes (*CsaPEBP1*/*2*/*5*/*7*) was performed from the published RNA-seq data of the corolla opening. Results showed that the expression patterns of *CsaPEBP1*, *CsaPEBP2*, *CsaPEBP5*, and *CsaPEBP7* were consistent with those of 66, 20, 50, and 23 TFs, respectively (Figure 8a, Appendix A), with correlation coefficients varying from 0.8 to 1.0 (Figure 8b). Further, the TFs with a strong positive correlation (correlation index > 0.99) were selected for screening interactors of CsaPEBPs (Appendix A).

Multiple TF binding sites (TFBSs) were identified in the promoter regions of the *CsaPEBP1*, *CsaPEBP2*, *CsaPEBP5*, and *CsaPEBP7* genes, which could correspond to the most TFs mentioned above (Figure 8c, Appendix A). Based on the sequence similarity, the TFBSs with 100% similarity to *Arabidopsis* ones were retained, as follows: one ERF, three NAC, and one HSF in the *CsaPEBP1* promoter; eight TALE in the *CsaPEBP2* promoter; two Dof, and five NF-YB TALE in the *CsaPEBP5* promoter; eight WRKY and 79 GATA in the *CsaPEBP7* promoter (Figure 8d). These elements provided critical evidence for whether TFs directly regulate *CsaPEBP* genes. Especially, the tandem repeats of NF-YB and GATA might enhance the binding abilities of Csa4G037610 and Csa7G064580 to the promoters of *CsaPEBP2* and *CsaPEBP5*, respectively, thereby regulating their expression.

## 3. Discussion

Proper flowering time for crops such as cucurbits means a greater yield and more robust adaptation [19]. Mining the vital genes related to flowering time would be beneficial to gain a better understanding of their genetic regulatory mechanisms, facilitating the cultivation of cucurbit varieties with superior growth periods and adaptations.

Comparative genomics analysis is acknowledged as a powerful approach to exploring the evolution and function of genes. In the current work, we examined the inter- and intra-specific differences between seven cucurbit species using a comparative genomics analysis of their *PEBP* genes. By integrating transcript expression profiles with promoter element prediction, these findings would empower us to discover the novel genes and their potential interactors involved in cucumber flowering.

### 3.1. Conservation of the PEBP Family in Seven Cucurbits

At the whole-genome level, eight *CsaPEBP*s, nine *CmePEBP*s, seven *ClaPEBP*s, 10 *CmoPEBP*s, 11 *CarPEBP*s, 10 *CpePEBP*s, and 10 *CmaPEBP*s were identified from cucumber, melon, watermelon, pumpkin, cushaw, zucchini, and winter squash, respectively. The number of PEBP family members in seven cucurbits was similar to *Arabidopsis* (7) but fewer than rice (19) and maize (25) [20,21]. This validated the theory that monocots had three to four times more PEBP family members than dicots did [22]. In actuality, cucurbits have a slightly higher number of *PEBP* genes than *Arabidopsis*, perhaps as a result of having a larger genome. Interestingly, out of the seven cucurbits, the tetraploid ones (pumpkin, cushaw, zucchini, and winter squash) have a larger number of *PEBP* genes than the diploid cucurbit ones (cucumber, melon, and watermelon), which is similar to the findings from Yi et al. [19].

According to *Arabidopsis*, a total of 65 PEBP proteins from seven cucurbits could be grouped into four main clades as follows: MFT, FT, TFL (including BFT-like, CEN-like, and TFL1-like subclades), and PEBP clades. Among these, the PEBPs from the MFT, FT, and TFL clades exhibited high sequence homology with *Arabidopsis* PEBPs, sharing the conserved PEBP domains and DPDxP motifs. Significantly, the MFT-like, FT-like, and TFL-like proteins may be distinguished by the 85th amino acid of AtFT, which explains why FT and TFL1 homologs functioned differently in several plants [22]. Consistently, the cucumber *FT* gene (*CsaPEBP2*) has been reported to be a key gene controlling early flowering, but *CsTFL1* (*CsaPEBP5*) inhibited terminal flower formation [14,15,23]. For the gene structure, most cucurbit *PEBP* genes contained four exons and had large variations in gene length, which might lead to the functional diversity of PEBP family members.

Gene duplication events are an important way for plant evolution to have high efficiency [12]. For the PEBP family in the seven cucurbits, whole-genome duplication might contribute more to gene diversity. Together with interspecies collinearity analysis and evolutionary relationships, the orthologous gene pairs with *Arabidopsis* were conducive to investigating the biological functions of *PEBP* genes in cucurbits. Notably, all of these gene pairs belonged to the MFT, TFL, and PEBP clades, indicating that they may have shared an ancestor with *Arabidopsis* and seven cucurbit crops. Moreover, the majority of collinear gene pairs belonged to one-to-one relationships, which was comparable to other crops, such as tomato [24]. Additionally, it was also noticed that the members from the PEBP clade, which encoded the PEBP domain present in bacteria and archaea, were distinct in both the conserved domain and gene structure, unlike PEBP members from other clades (Appendix A). Nevertheless, their phylogenetic relationships and conserved sequence regions provided evidence for the ancient common origin of a basic protein functional unit.

### 3.2. Functional Diversification of Cucumber PEBP Genes

Despite extensive sequence conservation, *PEBP* genes are well-known to be the key regulators of various biological processes, especially *FT*/*TFL-like* genes in the regulation of floral transition [1,25]. Considering that the exact molecular function of the cucurbit *PEBP* genes was, in most cases, still unclear, *CsaPEBP* genes from cucumber, a well-established model system for unisexual flower development, were chosen for expression pattern profiles. Both in RNA-seq and RT-qPCR analyses, eight *CsaPEBP* genes exhibited divergent expression levels in six tissues of cucumber plants. This resembled the reports on tomato, bamboo, and *Rosaceae* tree species [22,24,26]. More particularly, *CsaPEBP5*, *CsaPEBP6*, and *CsaPEBP7* in male flowers, as well as *CsaPEBP3* and *CsaPEBP4* in the stem, appeared to exhibit strong tissue-specific high expression. Even if the genes belonging to the same clade might function differently, such as *CsaPEBP1* and *CsaPEBP6* from the MFT-like subclade and *CsaPEBP5* and *CsaPEBP8* from the TFL-like subclade. For the genes expressed in either male or female flowers, *CsaPEBP1*, *CsaPEBP2*, *CsaPEBP5*, *CsaPEBP6*, and *CsaPEBP7* might act at the later stages of corolla opening. *CsaPEBP2* and *CsaPEBP5* have been reported to influence flowering in cucumber, namely *CsFT* and *CsTFL1* [14,15], and correspondingly, their expression trends were reversed during the female corolla opening process. Summarily, these variable expression patterns suggested functional diversification of *CsaPEBP* genes, which was supported by the variations in their gene structure.

### 3.3. Novel TFs That Potentially Interact with CsaPEBPs

In the plant flowering regulatory network, TFs generally function in conjunction with flowering-related genes to regulate flowering time. CO (Zinc finger), CIB1 (bHLH), and FLC (MADS-box) have been recognized as the classical TFs that directly bind to the FT and its homolog promoter and induce their expression [27], and recently several novel TFs have also been confirmed to do so. For example, *Arabidopsis* B3-Domain TF VAL1 regulates the floral transition by repressing FT [28]. *StABL1*, a TF central to abscisic acid signaling, binds to FT homologs to promote flowering and, consequently, a short life cycle in potatoes [29]. This demonstrated that other TFs also have the potential to directly interact with FT or its homologs. Considering this, a strategy combining clustering of time-series and promoter cis-element analysis was carried out. After screening, several TFs were obtained as possible roles for directly binding FT or other PEBPs during flowering response. Except for the ERF TF reported in *Arabidopsis*, there is still little information on the direct interaction between FT and the other TFs [30].

In addition, the tandem repeats of NF-YB and GATA binding sites occurred in the promoter regions of *CsaPEBP2* and *CsaPEBP5*, which may mean a stronger binding ability between TFs and CsaPEBPs. Although these findings need to be further verified, they can still provide novel evidence for understanding the flowering regulatory mechanism in cucurbit crops.

## 4. Materials and Methods

### 4.1. Search for PEBP Family Members in Cucurbits

Genome sequence data for seven cucurbit crops were retrieved from the Cucurbit Genomics Database (CuGenDB, http://cucurbitgenomics.org/, accessed on 21 April 2023), including cucumber (Chinese Long, v3), melon (DHL92, v3.6.1), watermelon (97103, v2), pumpkin (Rifu), cushaw (silver-seed gourd), zucchini, and winter squash (Rimu). Using these data, a Hidden Markov Model-based search was conducted to predict the PEBP proteins containing a conserved domain (PF01161). Also, seven *Arabidopsis* PEBP protein sequences from TAIR (https://www.arabidopsis.org/, accessed on 21 April 2023) were used as queries in BLASTp searches against the seven cucurbit genomes, respectively. Further, redundant protein sequences were eliminated using the InterPro database [31] and CD-Search tool [32].

### 4.2. Multiple Alignments, Phylogenetic and Synteny Relationships Analysis

Using the MUSCLE method with default settings, the PEBP family members from all seven cucurbit crops, as well as *Arabidopsis*, were sequenced by parameters. The outcome file was imported into the GeneDoc (version 2.7.0) software for visualization [33]. Based on the multiple sequence alignment, an NJ phylogenetic tree was conducted using MEGA 11 (version 11.0.13) software in accordance with the p-distance model, as well as 1000 bootstrapping replicates and other default options [34]. The NJ tree was displayed and optimized using an online tool iTOL [35].

Conserved syntenic blocks within and between the genomes of seven cucurbit species and *Arabidopsis* were identified and visualized using MCScanX (version 1.1) and TBtools-II (version 2.0) software [36,37].

### 4.3. Gene Localization and Structure Analysis

The chromosomal localization and exon/intron organization of the *PEBP* genes were investigated based on the GFF/GTF annotation files of seven cucurbit crop genomes from CuGenDB. The gene localization information was visualized using an advanced Circos plot in TBtools-II [37]. Then the exon number and length of each *PEBP* gene were counted.

### 4.4. Investigation of Cis-Elements in the Promoter Region

The 2000 bp DNA sequences upstream at the transcription start site (ATG) were extracted from the CuGenDB database as the promoter regions of *PEBP* genes. These sequences were uploaded to the PlantCARE database (http://bioinformatics.psb.ugent.be/webtools/plantcare/html/, accessed on 7 November 2023) to identify the putative cis-acting elements. To predict TFBSs, the FASTA file of the promoter sequences was submitted to the PlantPAN4.0 website (http://plantpan.itps.ncku.edu.tw/plantpan4/index.html, accessed on 19 February 2023) [38].

Also, the 7000 bp upstream regions of ATG were obtained using the above-mentioned method, which was used to analyze several unique *cis*-elements (e.g., CCAAT box and CORE1/2) [10]. The results were displayed using the TBtools-II (version 2.0) [37] and Illustrator of Biological Sequences (IBS) packages (version 1.0) [39], respectively.

### 4.5. Plant Sampling, RNA Preparation and RT-qPCR Analysis

The inbred cucumber line ‘D1008’ was used as the experimental material and was planted at a greenhouse in the Horticulture Experiment Station, Northeast Agricultural University (Harbin, China). The cultivation conditions were maintained at 28 °C/18 °C (day/night) and a 16 h day/8 h night cycle. At the fruiting stage, a total of five tissues (leaf, stem, tendril, female flower, and male flower) were separately collected. Otherwise, female and male flowers were sampled from 1, 3, 4, and 5 days after labeling (DAL; labeling was made when an ovary became visible), which represented the four stages of cucumber flowering: green bud, green-yellow bud, yellow bud, and flowering. Five individual plants were pooled to form one biological replicate, and three biological replicates were prepared for each tissue. All samples were stored at −80 °C for gene expression detection.

Extraction of total RNA was performed using the RNAprep Pure Plant Kit with DNase I (Tiangen, Beijing, China), and then the qualified RNA was used for the synthesis of first-strand cDNA using the HiScript 1st Strand cDNA Synthesis Kit (Vazyme, Nanjing, China). For each gene, the RT-qPCR protocol with cross-intron primers was executed at a qTOWER3 system (Analytik Jena, Jena, Germany) in triplicate (Appendix A). The cucumber *elongation factor 1 alpha* (*CsEF1α*) gene, as an internal control, was used to calculate the relative transcription levels of target genes using the 2^−ΔΔCt^ method. The statistical analysis of gene expression was performed by using SPSS software (Version 18.0).

### 4.6. Clustering of Time-Series and Correlation Analysis Based on RNA-Seq Data

The RNA-seq data published by Sun et al. [40] were acquired from the NCBI Gene Expression Omnibus database (GEO accession: GSE76358), which referred to the corollas at 1, 3, 4, and 5 DAL from the normal ovary. All protein sequences of the differentially expressed genes were submitted as queries to the PlantTFDB database [41] to screen the TFs. Also, the *CsPEBP* genes in this study were matched in the RNA-seq data using a BLASTp search. Finally, a total of 8 cucumber *PEBP* genes and 1251 *TF* genes were identified and shown in Appendix A. Gene expression levels were measured using transcripts per million (TPM) values.

Clustering of time-series analysis was performed using the Short Time-series Expression Miner (STEM) program [42], and the maximum number of model profiles was set to fifty.

Pearson correlation coefficients of expression levels between *CsPEBP*s and *TF*s were estimated using GraphPad Prism 9 and visualized as a radar chart.

## 5. Conclusions

In short, 65 *PEBP* genes were comparatively identified and characterized in seven cucurbit crops, including their evolutionary, structural conservation, and functional diversification. A distinct PEBP clade was noticed, encoding the conserved domain in bacteria and archaea. *CsaPEBP1* was highly expressed in male and female flowers, while *CsaPEBP5*, *CsaPEBP6*, and *CsaPEBP7* were active in cucumber males. All four genes might function at the later stages of corolla opening. Furthermore, several cucumber TFs were inferred to be partners of FT or other PEBPs; of these, Csa4G037610 and Csa7G64580 might have more vital binding ability due to the tandem repeats of NF-YB and GATA in *CsaPEBP2* and *CsaPEBP5* promoters. Together, these outcomes enrich our understanding of the primary characteristics of the PEBP family in cucurbit crops and provide novel thoughts for flowering regulation through TFs.

## Figures and Tables

**Figure 1 ijms-25-03815-f001:**
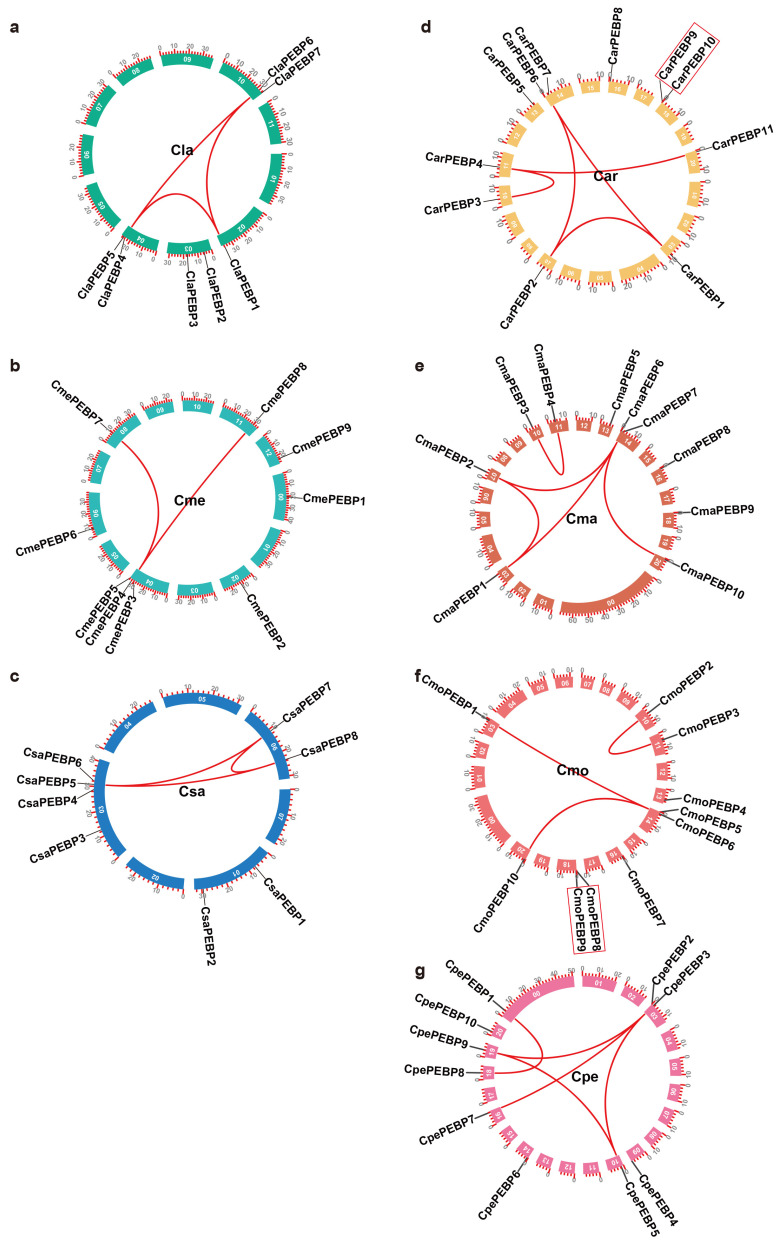
Localization and synteny of the *PEBP* genes in seven cucurbit crops. (**a**) *ClaPEBP*s in the watermelon genome. (**b**) *CmePEBP*s in the melon genome. (**c**) *CsaPEBP*s in the cucumber genome. (**d**) *CarPEBP*s in the cushaw genome. (**e**) *CmaPEBP*s in the winter squash genome. (**f**) *CmoPEBP*s in the pumpkin genome. (**g**) *CpePEBP*s in the zucchini genome. Gene pairs with a syntenic relationship were joined by the red lines. Tandem duplicated genes were laid in a red box.

**Figure 2 ijms-25-03815-f002:**
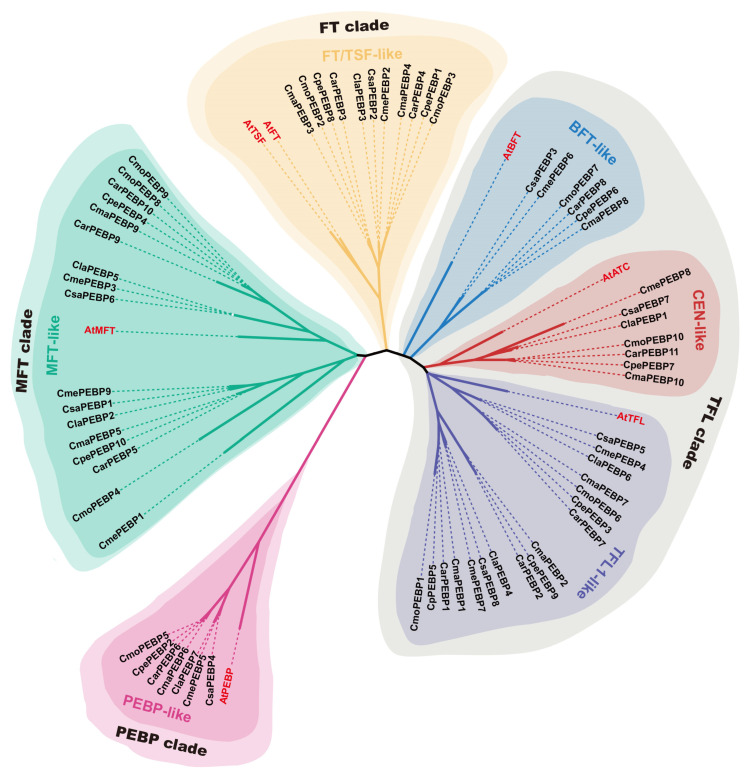
Evolutionary tree of FT/PEPB-related proteins in *Arabidopsis* and seven cucurbit crops.

**Figure 3 ijms-25-03815-f003:**
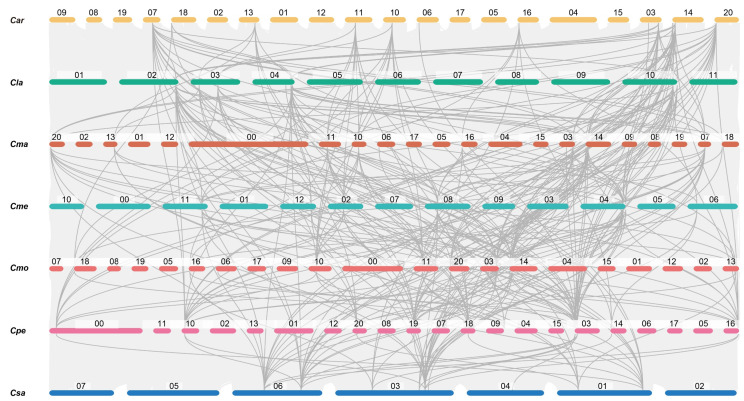
A syntenic map of *PEBP* genes among seven cucurbit crops.

**Figure 4 ijms-25-03815-f004:**
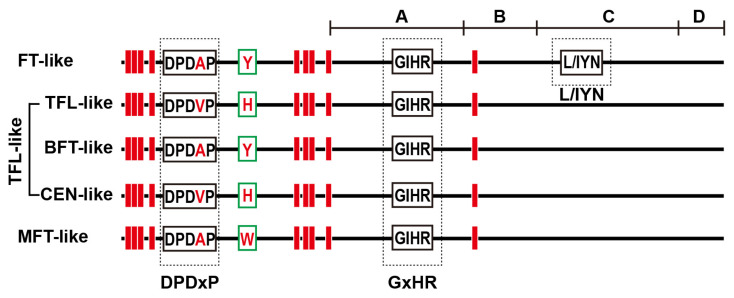
Pattern diagram of PEBP family proteins from seven cucurbit crops. Red rectangle indicates amino acid residues that interact with 14-3-3 protein. The green rectangle indicates a key amino acid residue that determines MFT-like, FT-like and TFL-like proteins. Black boxes represent the conserved DPDxP, GxHR motif and L/IYN. Underlines represent segments A, B, C and D, respectively.

**Figure 5 ijms-25-03815-f005:**
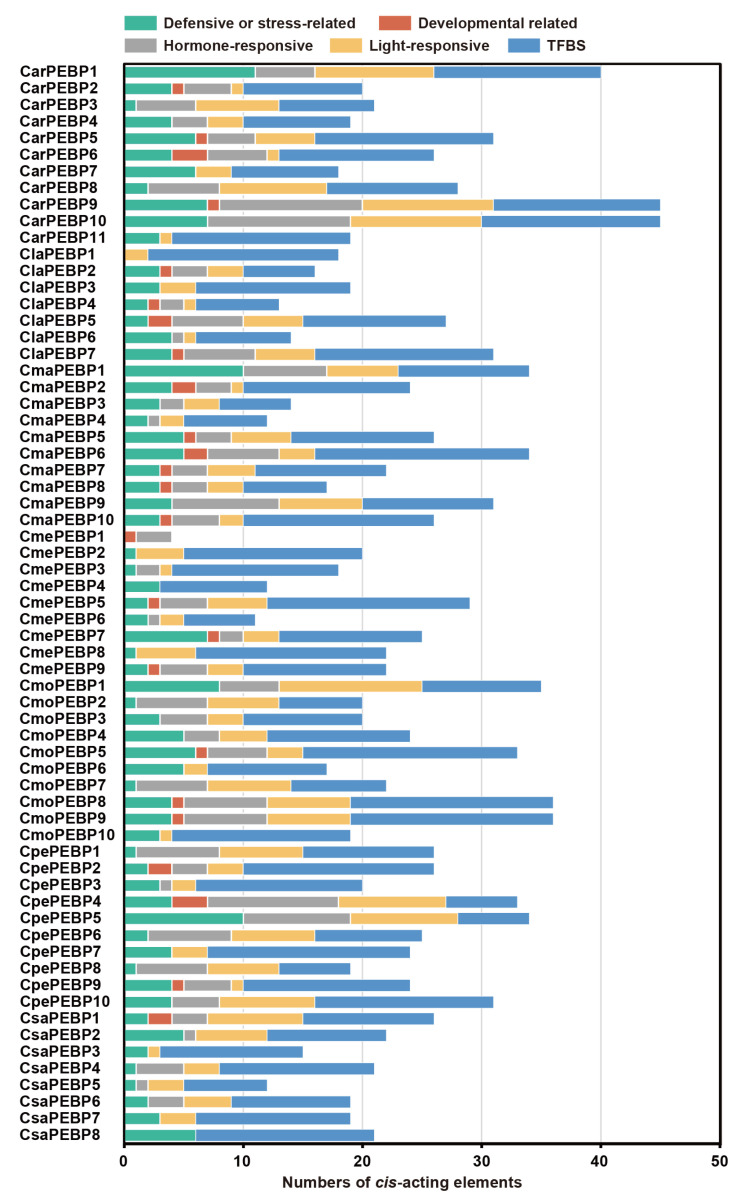
Prediction of *cis*-acting elements in the cucurbit *PEBP* gene promoters.

**Figure 6 ijms-25-03815-f006:**
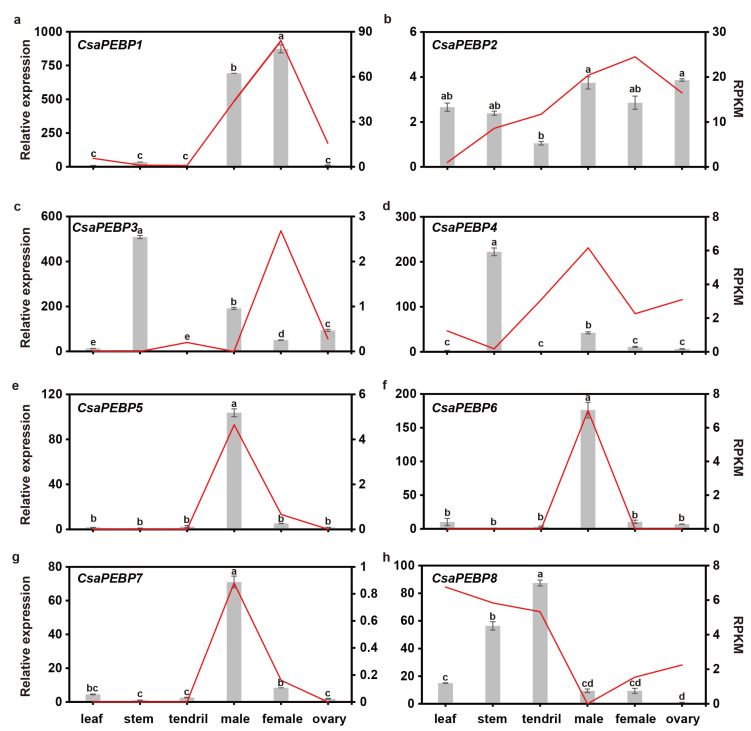
Expression profiles of *CsaPEBP1*–*CsaPEBP8* genes (**a**–**h**) in cucumber leaf, stem, tendril, male, female and ovary. Grey columns indicate the relative gene expression levels determined using RT-qPCR, and red lines represent the transcript abundance changes calculated using the Reads Per Kilobase per Million mapped reads (RPKM) method. Lowercase letters (for example, a, b, c, etc.) above the columns represent significant differences at the 0.05 level based on Tukey’s test.

**Figure 7 ijms-25-03815-f007:**
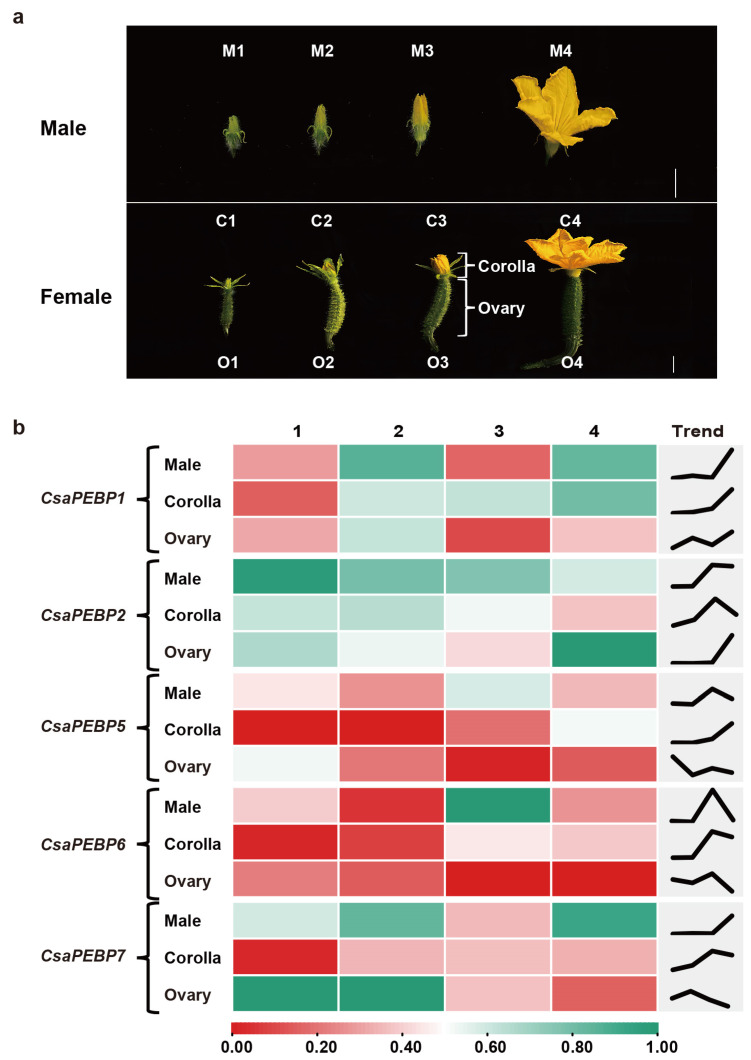
Expression patterns of *CsaPEBP* genes in the process of cucumber flowering. (**a**) Progression of corolla opening in the normal male (M; top) and female (C, corolla and O, ovary; bottom), including four stages: green bud (1), green-yellow bud (2), yellow bud (3), and flowering (4). Bar = 1 cm. (**b**) Heatmap (left) showing different expression levels of five *CsaPEBP* genes in male and female flowers at four developmental stages. Changes in gene expression levels are shown in color as the scale. Mini Line Chart (right) shows the mean expression trend of each gene.

**Figure 8 ijms-25-03815-f008:**
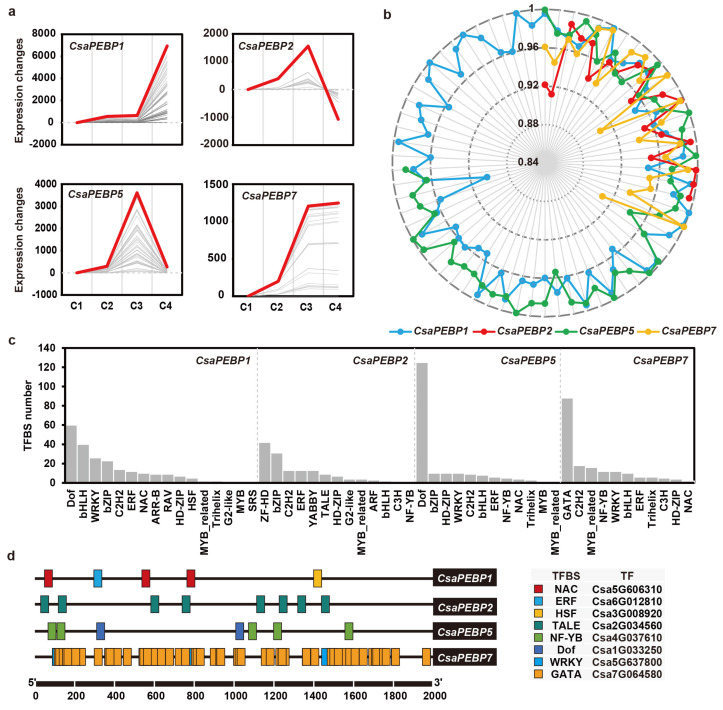
Screening for TFs that interact with CsaPEBPs. (**a**) Time-series clusters of *CsaPEBP*s and *TF*s based on RNA-seq data. The horizontal axis represents four developmental stages in the corollas of a female flower: green bud (C1), green-yellow bud (C2), yellow bud (C3), and flowering (C4). Trend lines are color-encoded with gray shades denoting high membership values of genes belonging to the time-series cluster, and the red lines indicate the expected trend. (**b**) Radar chart of the Pearson correlation coefficient between expression levels of *TF*s and *CsaPEBP*s. (**c**) Number of transcription factor binding sites (TFBSs) in *CsaPEBP* gene promoters. (**d**) Putative TFs interacting with CsaPEBPs.

## Data Availability

The original contributions presented in the study are included in the article.

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
