# Peer review of "Comparative Genomic Analysis of PEBP Genes in Cucurbits Explores the Interactors of Cucumber CsPEBPs Related to Flowering Time"

_ijms, 2024, doi:10.3390/ijms25073815_

Round 1
Reviewer 1 Report
Comments and Suggestions for Authors
In this work, a total of 65 PEBP genes were identified from seven cucurbit crops and were used to perform a comparative genomics analysis. Chromosomal locations, orthologous and paralogous relationships, protein-conserved domains, and promoter cis-elements among these PEBP genes were determined. Also, tissue-specific expression analysis was employed…
Generally, the manuscript is well-written and the work is worth publishing. However, the manuscript has some concerns before acceptance of the article for publication.
Line 2: The title is too long.
Line 21-29: It is better to present the results in the form of quantitative in the abstract not only descriptive
In general, the presentation is in the form of Figures in the entire results section. In contrast, statistical analysis in the form of Tables and quantitative values is rarely seen.
Line 30: Keywords should be in alphabetical order and different from title words.
Line 71-80: At the end of the introduction section, first, briefly mention what the scientific gap is in this field. And why did you have this goal? You wrote here the results of the work!!
Line 102-108: Is the more exact mapping position on the chromosomes available?
Line 113: Why use Neighbor-Joining To investigate the evolutionary relationships of the PEBP proteins? Was the UPGMA method also checked?
Line 205: Did you perform any statistical analysis to compare means of gene expression experiments?
Line 409: Did you check the efficiency of PCR because you used the Livak method? You used to calculate the relative transcription levels of target genes by the 2−ΔΔCt method
Comments on the Quality of English Language
Minor editing of English language required
Reviewer 2 Report
Comments and Suggestions for Authors
the manuscript need some improvements before final decision. some comments are:
1. in line 22 what is "active in cucumber male or female"? please make it clear.
2. line 23-25 not clear what author actually want to say?
3.all the gene names should be italic.
4. line 103-104 is not clear and hard to understand.
5. line 110-111, how you can conclude few genes disappear on chromosomes.
6. what is "Ulteriorly"?
7. line 120-122, too long and unable to understand. please re-write it.
8. remove term respectively from line 136.
9. remove discussion from results section
Comments on the Quality of English Language
lack punctuation, grammar mistakes
Reviewer 3 Report
Comments and Suggestions for Authors
Dear Authors,
The following manuscript was submitted to Molecular Science: A Comparative Genomic Analysis of PEBP Family Members in Cucurbits Explores the Interactors of Cucumber PEBP Proteins Involved in Regulating Flowering Time.
While the study is potentially interesting, the manuscript needs improvement. First, language corrections are required in many places. Also, some of the figures need improvement.
I have corrected the abstract.
Abstract:
The family of phosphatidylethanolamine-binding proteins (PEBPs) participates in various plant biological processes, mainly flowering regulation and seed germination. In cucurbit crops, several PEBP genes have been recognized to be responsible for flowering time. However, the investigation of PEBP family members across the genomes of cucurbit species has not been reported, and their conservation and divergence in structure and function remain largely unclear. Herein, PEBP genes were identified from seven cucurbit crops and used to perform a comparative genomics analysis. The cucurbit PEBP proteins could be classified into MFT, FT, TFL, and PEBP clades, and further, the TFL clade was divided into BFT-like, CEN-like, and TFL1-like subclades. The MFT-like, FT-like, and TFL-like proteins were clearly distinguished by a critical amino acid residue at the 85th Arabidopsis FT protein. Gene expression analysis revealed that CsaPEBP1, CsaPEBP2, CsaPEBP5, CsaPEBP6, and CsaPEBP7 were active in cucumber males or females and displayed the highest expression at the later stages of corolla opening. Together with promoter cis-element analysis, clustering of time-series-based RNA-seq data in cucumber corolla opening found potential interacting transcription factors (TFs) with four CsaPEBPs. Because of the tandem repeats of binding sites in promoters, NF-YB (Csa4G037610) and GATA (Csa7G64580) TFs appeared to be better able to regulate the CsaPEBP2 and CsaPEBP5 genes, respectively. This study would provide helpful information for further investigating the roles of PEBP genes and their interacting TFs in growth and development processes, such as flowering time regulation in cucurbit crops.
Keywords: Please do not use the exact words from the title here.
Introduction:
Lines 62-64 Please correct all the Latin names of the listed crops; botanical indications are missing, e.g., Cucumis sativus L.
Line 72: Start a new section.
.Lines 71-81: this section is more suitable for conclusion.
Please formulate a clear goal for this study.
Results
It is well-written and formulated.
Several figures are complicated to read and follow!
Please correct the Figures 4 and 5.
Discussion
It is well-written and structured. Several grammar mistakes need to be corrected in this section!
Correct formulation:
Line 273: Proper flowering time for crops such as cucurbits means a greater yield and more robust adaptation [19].
Material and Methods
It is well-written and structured.
Conclusions
I have corrected this section of the manuscript!
In short, 65 PEBP genes were comparatively identified and characterized in the seven cucurbit crops, including their evolutionary, structural conservation, and functional diversification. A distinct PEBP clade was noticed, encoding the conserved domain in bacteria and archaea. Expression profiles revealed that CsaPEBP1, CsaPEBP2, CsaPEBP5, CsaPEBP6, and CsaPEBP7 were active in cucumber males or females, with higher expression levels and might function at the later stages of corolla opening. Furthermore, several cucumber TFs were inferred to be partners of FT or other PEBPs; of these, Csa4G037610 and Csa7G64580 might have more vital binding ability due to the tandem repeats of NF-YB and GATA in CsaPEBP2 and CsaPEBP5 promoters. Together, these outcomes enrich our understanding of the primary characteristics of the PEBP family in cucurbit crops and provide novel thoughts for flowering regulation through TFs.
Overall, this manuscript needs several corrections before it can be published.
22.3.2024
Comments on the Quality of English Language
The manuscript needs language improvement; several grammar mistakes and syntax must be corrected.
